# Effectiveness of Pilates and Yoga to improve bone density in adult women: A systematic review and meta-analysis

Rubén Fernández-Rodríguez[1,2☯], Celia Alvarez-Bueno[2,3☯]*, Sara Reina-Gutiérrez[2‡], Ana Torres-Costoso[4☯], Sergio Nuñez de Arenas-Arroyo[2‡], Vicente Martínez-Vizcaíno[2,5☯]

**1** Movi-Fitness S.L, Universidad de Castilla La-Mancha, Cuenca, Spain, **2** Health and Social Research Center, Universidad de Castilla La-Mancha, Cuenca, Spain, **3** Universidad Politécnica y Artística del Paraguay, Asunción, Paraguay, **4** Escuela de Fisioterapia y Enfermería, Universidad de Castilla-La Mancha, Toledo, Spain, **5** Facultad de Ciencias de la Salud, Universidad Autónoma de Chile, Talca, Chile

☯ These authors contributed equally to this work.
‡ These authors also contributed equally to this work.
* celia.alvarezbueno@uclm.es

## Abstract

### Background

The ageing population brings about the appearance of age-related health disorders, such as osteoporosis or osteopenia. These disorders are associated with fragility fractures. The impact is greater among postmenopausal women due to an acceleration of bone mineral density (BMD) loss.

### Objective

To estimate the effectiveness of Pilates or Yoga on BMD in adult women.

### Methods

Five electronics databases were searched up to April 2021. Randomized controlled trials (RCTs), non-RCTs and pre-post studies were included. The main outcome was BMD. Risk of bias was evaluated using the Cochrane risk of bias tool. A random effects model was used to pool data from primary studies. Subgroup analyses based on the type of exercise were conducted.

### Results

Eleven studies including 591 participants aged between 45 and 78 years were included. The mean length of the interventions ranged from 12 to 32 weeks, and two studies were performed for a period of at least one year. The pooled effect size for the effect of the intervention (Pilates/Yoga) vs the control group was 0.07 (95% Confidence interval [CI]: -0.05 to 0.19; $I^2$ = 0.0%), and 0.10 (95% CI: 0.01 to 0.18; $I^2$ = 18.4%) for the secondary analysis of the pre-post intervention.

**Data Availability Statement:** All relevant data are within the paper and its Supporting Information files.

**Funding:** This study was funded by European Regional Development Fund. The funders had no role in study design, data collection and analysis, decision to publish, or preparation of the manuscript.

**Competing interests:** The authors have declared that no competing interests exist.

## Conclusions

Despite of the non-significant results, the BMD maintenance in the postmenopausal population, when BMD detrimental is expected, could be understood as a positive result added to the beneficial impact of Pilates-Yoga in multiple fracture risk factors, including but not limited to, strength and balance.

## Introduction

The prevalence of age-related bone health disorders such as osteoporosis or osteopenia are growing as the proportion of older adults increases [1]. These disorders are characterised by a deterioration of bone health indicators, such as bone mineral density (BMD) and bone mineral content (BMC) [2, 3], which in turn increase the risk of osteoporosis-related fractures [4]. Moreover, these fractures are associated higher mortality and morbidity in both men and women [5], although women may be at increased risk, specially postmenopausal women [6], who are particularly exposed to an accelerated BMD loss as a consequence of reduced estrogen production [3].

Concerning possible approaches to strengthen bone tissue, a pharmacological approach may improve bone mass, but it presents side effects [7], such as deleterious effects on bone quality and architecture resulting in further fragility [4]. In this context, non-pharmacological approaches, such as physical activity or exercise, have been proposed as both preventive and therapeutic strategies [5]. Increasing physical activity levels has been related with the preservation of BMD [8] and physical function, and consequently with a reduction in the risk of fracture [9]. Likewise, exercise interventions should specifically address bone remodelling [5], considering different patterns of mechanic stress.

Mind-body methods, such as Yoga and Pilates [10–12], are exercise modalities that have been recommended to improve bone health since they include balance postures, which are intended to decrease the risk of falls [13–15], as well as muscular strengthening, which induces improvements on BMD [5]. Despite of the combined classification in Mind-body techniques, Pilates and Yoga present differences that may have influence on bone. For instance, Pilates is a therapeutic exercise highly focused on core-strengthening while Yoga is more related to breathing and meditation exercises. However, evidence about the beneficial effect of these exercise modalities is still controversial. While Yoga has been independently associated with a reduction in the risk of lower limb and hip fracture among postmenopausal women [16], and several authors have suggested improvements in BMD after Pilates training [17, 18], other studies have not observed changes following Pilates or Yoga interventions [19, 20], concluding that the stimulus caused by these exercise modalities are not appropriate for the bone remodelling process.

Since a study synthesizing the growing evidence in this field is missing, the aim of the present systematic review and meta-analysis was to estimate the effectiveness of Pilates or Yoga on the improvement or maintenance of bone health in adult women. Additionally, the study aimed to explore whether the effects of Pilates or Yoga depend on menopausal status, the type of intervention (Pilates vs Yoga), participants' mean age or baseline BMD values.

## Methods

### Search strategy and study selection

The present review and meta-analysis was conducted based on the recommendations of the Cochrane Handbook for Systematic Reviews of Interventions [21], and reported following the

Preferred Reporting Items for Systematic Reviews and Meta-Analyses (PRISMA) [22]. This review was registered in the PROSPERO database (registration number: CRD42020157143).

A systematic search in MEDLINE (via PubMed), Embase (via Scopus), CINAHL, the Physiotherapy Evidence Database (PEDro) and the Cochrane Central Register of Controlled Trials was conducted from inception until April 2021 for studies that aimed to determine the effectiveness of Pilates or Yoga on BMD among adult women. The search strategy was conducted combining Medical Subject Headings, free-terms and matching synonyms, including the following words: (1) population: adult, elderly, menopausal, postmenopausal, premenopausal; (2) intervention: Pilates, mind-body, 'exercise movement techniques', Yoga; (3) and outcome: 'bone mineral density', 'bone health', 'bone mineral content', 'T-score', DXA. Additionally, the references included in the identified publications deemed eligible were screened. The search strategy for MEDLINE is displayed in S1 Table.

## Eligibility criteria

Two independent reviewers (R. F.-R. and C. A.-B.) examined the titles and abstracts of retrieved articles to identify potentially eligible studies. The studies in which the titles and abstracts were related to the purpose of the present review were selected for full text screening. Inclusion criteria were: (1) type of studies: randomized controlled trials (RCTs), non-RCTs or pre-post studies; (2) type of participants: adult women with a mean age ≥ 45 years and in any menstrual status (premenopausal, postmenopausal); and (4) type of intervention: mind-body exercises based on 'Pilates' or 'Yoga' principles. Moreover, the studies were excluded when: (1) outcome measurements were not reported as BMD or T-score values, or (2) the data to calculate effect size (ES) estimates were not available. In cases of initial disagreement between reviewers, a third reviewer (V. M.-V.) consulted. No language restrictions were applied.

## Data extraction and risk of bias assessment

Two reviewers (R. F.-R. and C. A.-B.) independently extracted the following information from the included studies: first author's name and year of publication; study design; characteristics of the participants (premenopausal/postmenopausal); mean age; sample size; weekly frequency and length of the Pilates or the Yoga intervention; the reported BMD and T-score values, and the main results of each study. In cases of initial disagreement between reviewers, a third reviewer (V. M.-V.) was consulted.

The Cochrane risk-of-bias tool for randomized trials (RoB 2.0) [23] was used to assess the risk of bias of the studies included. The following domains were assessed: randomization process, deviations from intended interventions, missing outcome data, measurement of the outcome, and selection of the reported result. Each domain was assessed for risk of bias following the instructions reported by the RoB 2.0 tool that provide a 'low risk of bias', 'some concerns' or 'high risk of bias' classification [24]. Accordingly, the overall risk of bias for each study was classified as (1) 'low risk of bias' when a low risk of bias was determined for all domains; (2) 'some concerns' when at least one domain was assessed as raising some concerns, but not to be at high risk of bias for any single domain; or (3) 'high risk of bias' when high risk of bias was reached for at least one domain or some concerns in multiple domains [23].

Non-RCTs and pre-post studies were assessed using the Quality Assessment Tool for Quantitative Studies [25], in which seven domains were evaluated: selection bias, study design, confounders, blinding, data collection methods, withdrawals and dropouts. Each domain was considered strong, moderate, or weak. Studies were categorised as (1) 'low risk of bias' when no weak ratings were present; (2) 'moderate risk of bias' when there was at least one weak rating; or (3) 'high risk of bias' when there were two or more weak ratings [25].

Risk of bias was independently assessed by two reviewers (R. F.-R. and C. A.-B.). A third reviewer (V. M.-V.) was consulted in case of disagreement.

## Data analysis

Primary data from each study was extracted, including mean BMD and T-score values, standard deviation of pre-post intervention and sample size. ES and related 95% confidence intervals (CIs) were calculated for each study [26]. The DerSimonian and Laird random effects method [27] was used to compute pooled ES estimates and respective 95% CIs. The pooled ES for the effect of Pilates/Yoga intervention vs the control group (CG) was estimated. Likewise, in order to show a meaningfully picture of the available evidence, an additional analysis based on the pre-post effect of Pilates/Yoga on the intervention group was conducted. Heterogeneity was evaluated using the $I^2$ statistic, with $I^2$ values of 0% - 40% considered to be 'not important' heterogeneity; 30% - 60% representing 'moderate' heterogeneity; 50% - 90% representing 'substantial' heterogeneity, and 75% - 100% representing 'considerable' heterogeneity [21]. The corresponding p-values and 95% CIs were also considered for the assessment of $I^2$ heterogeneity [28].

Furthermore, a sensitivity analysis was conducted to determine the robustness of the summary estimates by removing each included study one by one. Moreover, studies conducted in premenopausal or 'not specified' menstrual status women were removed in order to estimate the pooled ES for the effect of the Pilates/Yoga intervention among postmenopausal women. Additionally, subgroup analyses based on the type of intervention (Pilates vs Yoga), length (≤ 24 weeks or >24 weeks) and menopausal status as well as meta-regression models by mean age, baseline BMD values after adjusting for height and baseline body mass index (BMI) and length were conducted to determine their potential effect on the pooled ES estimates. Finally, the publication bias was evaluated through visual inspection of funnel plots and Egger's regression asymmetry test for the assessment of small study effects [29]. All statistical analyses were performed using StataSE v. 15 (StataCorp, College Station, TX, USA).

## Results

### Study selection

Eighteen potential studies were identified after the screening of titles and abstracts. Following the full text review of suitable articles, 11 studies [17–19, 30–37] were included in the present systematic review and meta-analysis, as five did not report the outcomes of interest, one did not include the population of interest and one did not have an intervention design. Further details are presented in Fig 1.

### Characteristics of studies

Characteristics of the studies and the interventions are summarized in Table 1. Among the 11 studies included, five were RCTs (41.7%) [17, 18, 30, 34, 35], four were pre-post studies (41.7%) [19, 31–33] and two were non-RCTs (16.6%) [36, 37].

### Participants

All included studies were conducted between 2009 and 2018, with a total of 591 participants, of which, 458 were in the intervention groups (77.5%): 150 in the Pilates groups (32.8%) and 308 in the Yoga groups (67.2%), and 133 in the CG (22.5%). Considering all participants, 535 were categorised as postmenopausal (90.5%), 34 as premenopausal (5.8%) and 22 as adult women (3.7%) because the study did not report menstrual status.[19] The mean age of

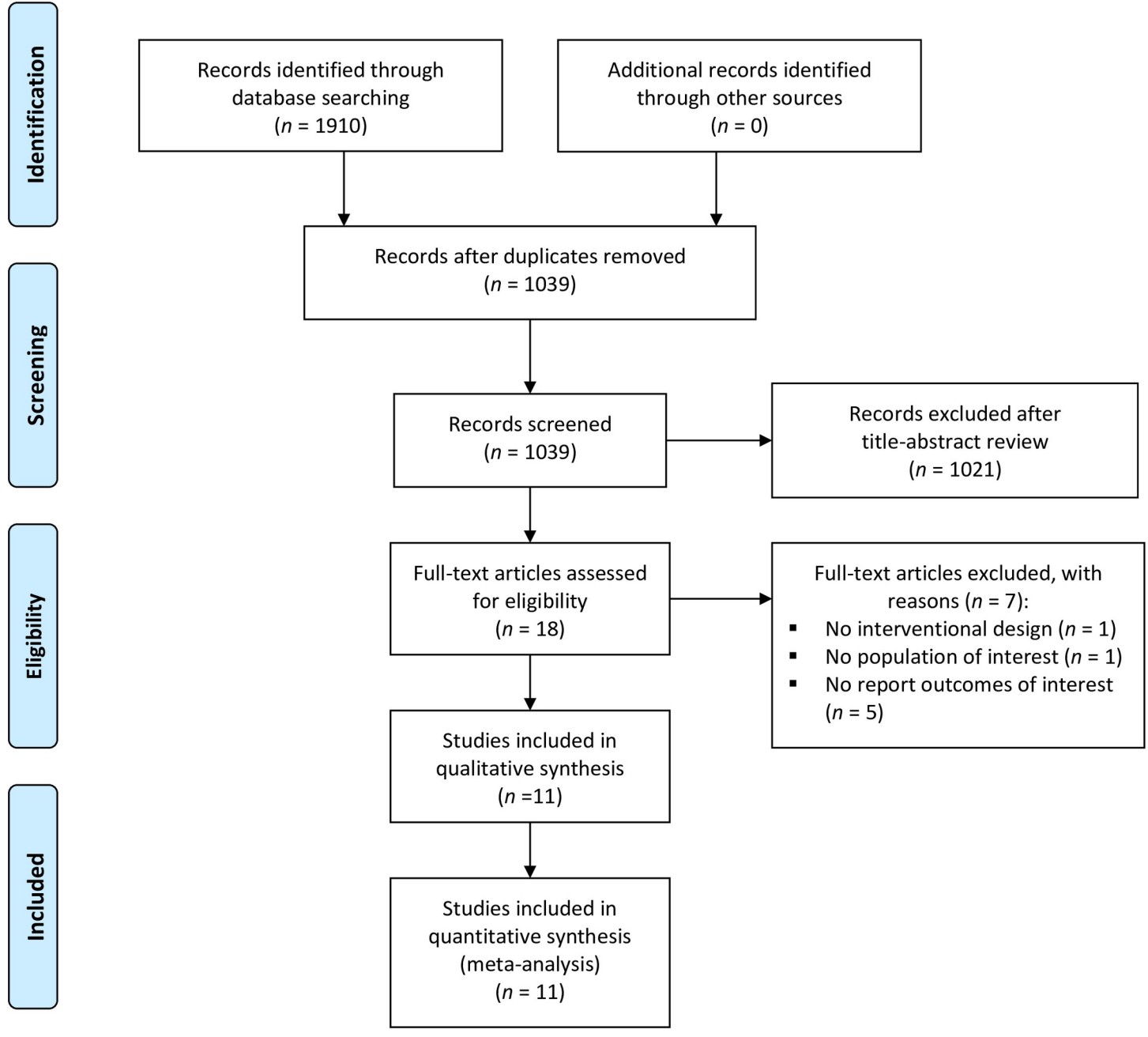

**Fig 1. Flow of the included studies.**

participants ranged from 45 to 78 years and their BMI from 21.1 to 27.6 kg/m². Further details are presented in Table 1.

## Interventions

Participants in the CG usually performed no activity or maintained their current physical activities without a specific exercise prescription [17, 18, 30, 34, 35]. Regarding the main characteristics of the interventions, five studies were conducted using the Pilates method and six

**Table 1. Characteristics of the included studies.**

| Reference | Design | Participants' characteristics | Mean Age | Sample size | Exercise | Frequency (s/wk) | Period | Outcome measure | Outcome results |
|---|---|---|---|---|---|---|---|---|---|
| Irez et al, 2009 [34] | RCT | Elderly females | IG: 72.8 ±6.7 CG: 78.0 ±5.7 | n = 60 IG: 30 CG: 30 | Pilates | 60'; 3s/wk | 12wks | DXA scan (Lunar DPX-IQ, Lunar Corp., Madison, WT): L2-L4 BMD and T-score; Femur BMD and T-score (gr/cm$^2$). Pre-post and 1 year follow up. | After one year of follow-up prominent decreases on BMD in the CG. |
| Bezerra et al, 2010 [30] | RCT | Postmenopausal | IG: 63.9 ±5.7 CG: 65.3 ±3.9 | n = 48 IG: 24 CG: 24 | Yoga | 60'; 3s/wk | 24wks | DXA scan (Lunar DPX-IQ, Lunar Corporation, Madison, WI): total body, lumbar spine, femoral neck, Ward's triangle, trochanter, total hip and forearm (gr/cm$^2$). | Spinal lumbar and total hip BMD decreased in CG (p<0.05), only spinal lumbar BMD decreased in IG |
| Kang et al, 2014 [36] | Non controlled CT | Postmenopausal | IG: 76.8 ±4.4 | IG: 11 | Yoga | 60'; 3s/wk | 12wks | DXA scan (QDR-4500, Hologic Inc., Waltham, MA, USA): BMD lumbar spine and BMC (gr/cm$^2$). | No significantly changes. |
| Angin et al, 2015 [17] | RCT | Postmenopausal Osteoporosis | IG: 58.2 ±5.5 CG: 55.9 ±9.2 | n = 41 IG: 22 CG: 19 | Pilates | 60'; 3s/wk | 24wks | DXA scan (Norland XR- 800 Densitometer Machine): T-score values L2-L4, BMD (gr/cm$^2$). | BMD increased in the IG and decreased in the CG significantly (p<0.05) |
| Kim et al, 2015 [35] | RCT | Premenopausal | IG: 45.7 ±1.0 CG: 43.2 ±1.0 | n = 34 IG: 27– 16 CG: 20– 18 | Ashtanga-based Yoga | 60'; 2s/wk | 32wks | DXA scan (GE Lunar Prodigy, GE Medical Systems, encore 2002 Software v. 10.50.086): aBMD (total body, lumbar spine, proximal femur and tibia bone) (gr/cm$^2$). | Yoga did not increase significantly aBMD or tibia bone characteristics. |
| Mikalacki et al, 2015 [19] | Pre-post | Adult women (not specified) | IG: 48.2 ±9.6 | IG: 22 | Pilates | 45'; 3s/wk | 24wks | Sahara ultrasound bone Densitometer (Hologic, Inc., MA, USA): BMD (gr/cm$^2$) was estimated from BUA and SOS parameters. | BMD increased not significantly. |
| Aguado-Henche et al, 2016 [33] | Pre-post | Postmenopausal | IG: 67.9 ±7.3 | IG: 37 | Pilates | 60'; 2s/wk | 36wks | DXA scan (Norland XR- 26 Densitometer Machine): BMD L2-L4, BMD (gr/cm$^2$). | BMD increased in the IG significantly (p<0.05) |
| Lu et al, 2016 [32] | Pilot Pre-post | Postmenopausal | IG: 68.2 ±na | IG: 227 | Yoga | 12'; daily | 10 years | DXA scan: spine, hip and femur (gr/cm$^2$). | BMD improved spine, hips and femur (p = 0.05) |
| Motorwala et al, 2016 [31] | Pre-post | Postmenopausal | IG: 53.4 ±4.2 | IG: 30 | Yoga | 60'; 4s/wk | 24wks | DXA scan (Inbody, Maltron, Tanita): lumbar spine (gr/cm$^2$). | Improvement in T-score of DXA scan of -2.55 ±0.25 (post) vs -2.69 ±0.17(pre) |
| Şerbescu et al, 2017 [37] | Non-RCT | Postmenopausal | IG: 56.5 ±6.3 CG: 56.9 ±3.4 | n = 47 IG: 22 CG: 25 | Pilates | 60'; 2s/wk | 1 year | OsteoSysSonost 3000 device: BMD-T-score. | Bone parameters showed significant differences favouring the IG (p<0.01) |
| Oliveira et al, 2018 [18] | RCT | Postmenopausal | IG: 55.6 ±6.8 CG: 54.1 ±5.3 | n = 34 IG: 17 CG: 17 | Pilates | Na; 3s/wk | 24wks | DXA scan (Hologic QDR 1000 Plus, Waltham, Massachusetts): aBMD (lumbar spine, femoral neck, total hip, trochanter, intertrochanter and ward's area) (gr/cm$^2$). | BMD increased in the IG vs CG for the lumbar spine and trochanter (p≤0.05) |

s/wk: sessions per week; DXA: dual-energy x-ray absorptiometry; BMD: Bone mineral density; Na: Not available; IG: intervention group; CG: control group; BUA: Broadband ultrasound attenuation; SOS: Speed of sound.

according to Yoga principles. The mean frequency of training sessions ranged from two to four sessions per week. Moreover, the length of sessions varied from 45 to 60 minutes, though one study did not report time [18]. Finally, the length of the intervention lasted from 12 to 32 weeks, and two studies were performed for a period of at least one year [32, 37] (Table 1).

## Outcomes

Bone health was assessed through BMD (g/cm$^2$) or T-score values, which refer to the normalised scale for BMD in standard deviations related with a young healthy sex- and race-matched population [38]. The assessment of these outcomes was performed through DXA scans [17, 18, 30–36] or ultrasound bone densitometer devices [19, 37] (Table 1). Most BMD measures were from lumbar spine or hip and trochanter area, due to their importance on osteoporosis-related fractures.

## Risk of bias

After assessing the risk of bias of RCTs with the Cochrane Collaboration tool (RoB 2.0) [23], two RCTs (40%) were assessed as 'high risk' of bias and three (60%) as 'some concerns' in the overall risk of bias (S1 Fig). The 'Quality Assessment Tool for Quantitative Studies' [25] was used to assess the methodological quality of non-RCTs and pre-post studies, resulting in two studies (33%) scored as 'weak' and four (67%) as 'moderate' risk of bias (S2 Fig).

## Data synthesis

**Meta-analysis.** The pooled ES for the effect of the Pilates/Yoga intervention vs the CG was 0.07 (95% CI: -0.05 to 0.19; I$^2$ = 0.0%) (Fig 2). In the additional analysis, the pooled ES for the effect of the pre-post Pilates/Yoga intervention was 0.10 (95% CI: 0.01 to 0.18; I$^2$ = 18.4%) (Fig 3).

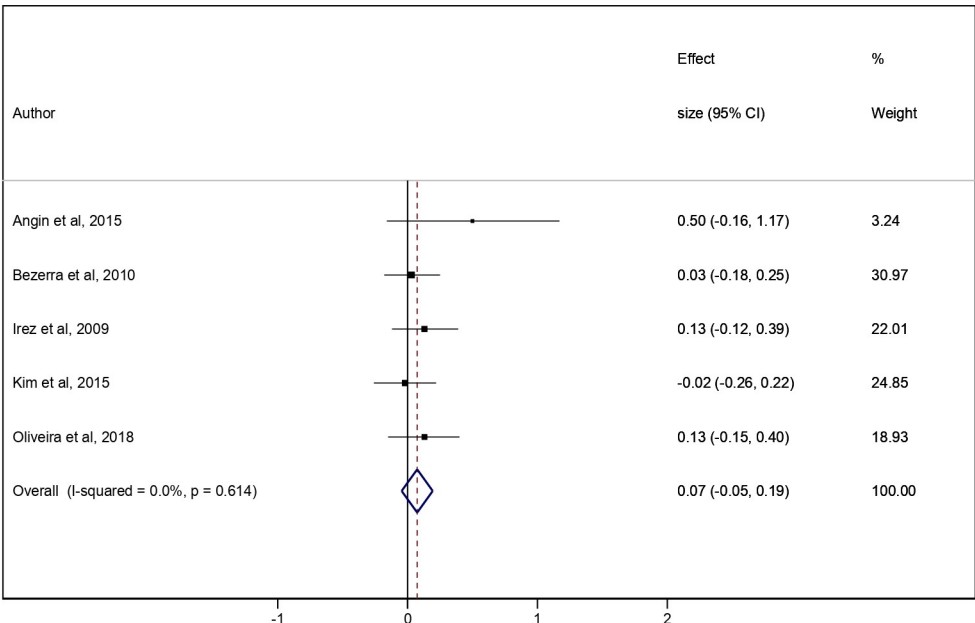

**Fig 2. Meta-analysis for intervention (Pilates and Yoga) vs the CG.**

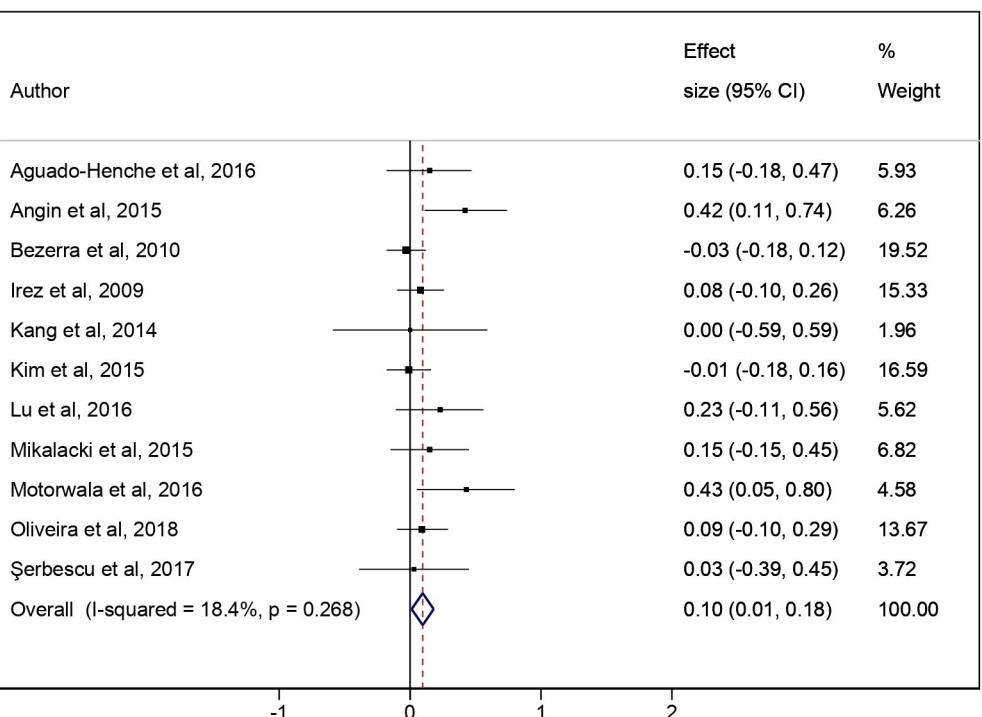

| Author | Effect size (95% CI) | % Weight |
|---|---|---|
| Aguado-Henche et al, 2016 | 0.15 (-0.18, 0.47) | 5.93 |
| Angin et al, 2015 | 0.42 (0.11, 0.74) | 6.26 |
| Bezerra et al, 2010 | -0.03 (-0.18, 0.12) | 19.52 |
| Irez et al, 2009 | 0.08 (-0.10, 0.26) | 15.33 |
| Kang et al, 2014 | 0.00 (-0.59, 0.59) | 1.96 |
| Kim et al, 2015 | -0.01 (-0.18, 0.16) | 16.59 |
| Lu et al, 2016 | 0.23 (-0.11, 0.56) | 5.62 |
| Mikalacki et al, 2015 | 0.15 (-0.15, 0.45) | 6.82 |
| Motorwala et al, 2016 | 0.43 (0.05, 0.80) | 4.58 |
| Oliveira et al, 2018 | 0.09 (-0.10, 0.29) | 13.67 |
| Şerbescu et al, 2017 | 0.03 (-0.39, 0.45) | 3.72 |
| Overall (I-squared = 18.4%, p = 0.268) | 0.10 (0.01, 0.18) | 100.00 |

**Fig 3. Meta-analysis for pre-post intervention (Pilates and Yoga).** Additional analysis based on the pre-post effect of Pilates/Yoga on the intervention group.

**Sensitivity analyses.** After studies were removed one at time from the analyses, none of them modified the pooled ES estimate (S2 Table). Additionally, when studies conducted in pre-menopausal [35] or 'not specified' menstrual status women [19] were removed, the results were not different for the Pilates/Yoga interventions vs the CG, nor for the pre-post Pilates/Yoga intervention analysis. Further details are available in S3 and S4 Figs.

**Subgroup analyses and meta-regression.** The subgroup analyses based on the type of exercise (Pilates or Yoga) compared with the CG showed that the pooled ES for the Pilates interventions was 0.16 (95% CI: -0.02 to 0.34; $I^2$ = 0.0%) while the pooled ES for Yoga was 0.01 (95% CI: -0.15 to 0.17; $I^2$ = 0.0%). Additionally, in the pre-post intervention analysis, the pooled ES for the Pilates intervention was 0.14 (95% CI: 0.03 to 0.25; $I^2$ = 0.0%) and, 0.06 (95% CI: -0.07 to 0.18; $I^2$ = 24%) for Yoga. Further details are available in S5 and S6 Figs. The subgroup analysis based on the length, ≤ 24 weeks or >24 weeks, was conducted in the pre-post intervention showing a pooled ES of 0.13 (95% CI: 0.00 to 0.25; $I^2$ = 40.5%) and 0.06 (95% CI: -0.07 to 0.19; $I^2$ = 0.0%), respectively (S7 Fig). Lastly, meta-analyses for interventions vs CG and for the pre-post intervention analysis according to menopausal status are available in S8 and S9 Figs.

The random-effects meta-regression models conducted based on age were not significant (p = 0.51) (S3 Table), neither were the meta-regression models based on baseline BMD values after adjusting for height (p = 0.36) or those based on BMI values (p = 0.45) (S4 Table) or length of the intervention (p = 0.57) (S10 Fig).

**Publication bias.** Publication bias was not observed, as evidenced by both funnel plot asymmetry and Egger's test (S5 Table).

## Discussion

The main purpose of our systematic review and meta-analysis was to estimate the effect of Pilates and Yoga interventions on BMD among adult women. Our findings showed that both Pilates and Yoga did not significantly improve BMD in adult women when compared with the CG. Considering only the intervention group analyses, a small significant improvement on BMD was found, particularly, for Pilates exercise interventions, and among postmenopausal women. Meta-regression and subgroup analyses showed that the results were not substantially modified by age, baseline BMD adjusted for height or BMI values, or length.

Our pre-post intervention results are in accordance with a previous Cochrane review that supports a small but statistically significant effect of exercise on BMD in postmenopausal women [39]. However, other studies have not shown the effect of physical exercise on BMD [19, 20, 40–42]. Despite these discrepancies, participating in regular exercise should be considered for a particularly exposed population at risk of osteoporosis, such as postmenopausal women, due to the benefits on the maintenance of a bone health indicator, such as BMD, and also because of the lack of side effects observed during exercise [39]. Finally, because the type of exercise may modify the effect on BMD [43], multicomponent strength and balance trainings have been recommended [39, 44] for improving not only bone health, but also physical function in daily life activities, and for preventing falls and osteoporosis-related fractures associated with the decline of BMD [45–47].

We found a small effect of the mind-body approach through Pilates and Yoga exercises to maintain BMD in postmenopausal women in the pre-post intervention analysis. However, we cannot ignore the fact that the evidence is not solid, and some studies reported non-significant differences on BMD through Pilates [19] or Yoga interventions [20, 35, 36]. The small sample size of most studies and the inadequate intervention length to produce adaptations in bone tissue [20, 36] are the main reasons for the weakness of the evidence. In this sense, it is well established that physical training should be maintained for at least one year to demonstrate substantial benefits in bone mass since the physiological cycle of bone remodelling lasts between four and six months [48], and only two of the studies included in this review accomplished this [32, 37].

In line with a recent systematic review estimating the effectiveness of exercise interventions for managing low bone mass in the forearm [8], our sensitivity analyses suggest a greater effectiveness of such interventions in postmenopausal than in premenopausal women. However, this comparison between postmenopausal and premenopausal women should be cautiously interpreted because there were only two studies conducted in premenopausal or 'not specified' menstrual status women [19, 35]. Apart from this, our subgroup analysis based on the type of exercise favoured Pilates instead of Yoga exercises. Two of the studies that did not show substantial changes on BMD were conducted using Yoga principles [35, 36]; however, as previously mentioned, it seems that the length of these interventions (12 weeks) was not long enough to accomplish adaptations in bone tissue. Finally, our results reinforced that exercise characteristics, such as type, length and intensity of exercise interventions, are key factors to induce changes on BMD [45], independently of the population characteristics. In this sense, it is supposed that high volume trainings lead to a smaller decrease of BMD in postmenopausal women [7] and also that the level of strain and body position during each exercise task may affect the load of the exercise impacting on BMD [47], but these factors cannot be addressed in our study.

Several mechanisms have been proposed to explain the benefits of physical exercise on BMD. One of the most accepted mechanisms is the increase of the vascular supplies to bone tissue as well as the angiogenic-osteogenic responses to exercise [49]. Furthermore, it is well

known that exercise causes mechanical stress on bone that may induce osteogenic effects [12, 43, 50]. Considering this, although both, Pilates and Yoga, are mind-body interventions, the physical demands on the musculoskeletal system varies across them. Pilates needs a specific co-contraction of the lumbo-pelvic and trunk stabiliser muscles that may produce forces on the spine, and the strengthening of this musculature may correlate with the density of bone [48]. This may explain why our data suggest a small significant improvement on BMD from Pilates, which was not found for Yoga, that includes several types of exercises, in which breathing or meditation techniques are the main components. As afore-mentioned and in line with our data, it seems that body position and physical demands during Pilates' exercises produced more mechanical stress on bone when compared with Yoga exercises.

Our study presents some limitations that should be stated. First, we should consider the risk of bias of the studies assessed. Second, the intensity of Pilates or Yoga interventions was not considered in our analyses since most studies did not report this information. Third, it was also not possible to take into account the exact time since menopause, which is intimately related with estrogen levels and BMD loss among adult women [11]. Fourth, drugs or dietary supplement intakes were not considered since three studies [18, 20, 33] reported that these co-interventions were not allowed. Fifth, as has been previously discussed, the length of the exercise intervention seems to be crucial in order to obtain effects on bone tissue due to the length of the physiological cycle of the bone remodelling process. Finally, other potential moderators, such as lean or fat mass, daily physical activity behaviours or diet, were not considered in our analyses, mainly due to the lack of information in the studies included.

Despite this, our study also presents some strengths: (1) we conducted an additional analysis based on the pre-post effects on the intervention group to show a meaningful picture of the available evidence; (2) the heterogeneity of results were categorised as 'not important'; and (3) the subgroup analyses and meta-regressions were conducted to control for potential sources of heterogeneity and bias.

## Implications for practice

Despite of the non-significant improvement on BMD after Pilates and Yoga interventions, these findings have occupational and public health implications that should be stated. For instance, public health policies may promote long-term physical exercise programs that could be based on Pilates or Yoga exercises to provide physical, social and psychological benefits that encourage active aging and self-management, as a part of a public health strategy to prevent possible risk factors associated with aging in women, without a negative impact on bone health [51].

## Conclusion

Our results suggest that mind-body exercises, such as Pilates and Yoga, did not produce a significant improvement on BMD among adult women when compared with the control groups. The multicomponent nature of Pilates and Yoga interventions, which include balance training and muscular strengthening in several weight-bearing postures, might be beneficial to improve multiple fracture risk factors in a clearly exposed population, such as postmenopausal women, thus, despite there were non-significant results, the maintenance of BMD should be considered as a positive result for this population. Lastly, we should consider that due to the short duration of the interventions and the small sample size of the conducted studies, additional randomized clinical trials specifically designed to improved bone health outcomes are needed to overcome the limitations described.

## Supporting information

**S1 Fig. Quality assessment for RCTs (RoB 2.0).** Green circles: low risk of bias; yellow circles: some concerns; red circles: high risk of bias.
(PDF)

**S2 Fig. Quality assessment for non-RCTs and pre-post studies.** Green circles: strong score; yellow circles: moderate score; red circles: weak score.
(PDF)

**S3 Fig. Meta-analysis for the intervention group vs CG among postmenopausal women.**
(PDF)

**S4 Fig. Meta-analysis for the pre-post intervention group among postmenopausal women.**
(PDF)

**S5 Fig. Meta-analysis for the intervention group vs CG by exercise (Pilates vs Yoga).**
(PDF)

**S6 Fig. Meta-analysis for the pre-post intervention group by exercise (Pilates vs Yoga).**
(PDF)

**S7 Fig. Meta-analysis for the pre-post intervention group by length ($\leq$ 24 weeks or $>$24 weeks.**
(PDF)

**S8 Fig. Meta-analysis for the intervention group vs CG by menopausal status.**
(PDF)

**S9 Fig. Meta-analysis for the pre-post intervention group by menopausal status.**
(PDF)

**S10 Fig. Meta-regression by length of the intervention.** ES: Effect size; Coef: coefficient; CI: confidence interval.
(PDF)

**S1 Table. Search strategy for the MEDLINE database.**
(DOCX)

**S2 Table. Sensitivity analyses.** ES: Effect Size; 95% CI: Confidence interval.
(DOCX)

**S3 Table. Meta-regression analyses by age.** [a]Significant at $p \leq 0.05$.
(DOCX)

**S4 Table. Meta-regression analyses by baseline BMD values after adjusting for height and for BMI.** BMD: Bone mineral density; BMI: Body mass index. [a]Significant at $p \leq 0.05$.
(DOCX)

**S5 Table. Publication bias by Egger's test.** [a]Significant at $p \leq 0.1$.
(DOCX)

**S1 File. PRISMA checklist.**
(DOCX)

## Author Contributions

**Conceptualization:** Rubén Fernández-Rodríguez, Celia Alvarez-Bueno, Ana Torres-Costoso.

**Data curation:** Rubén Fernández-Rodríguez, Celia Alvarez-Bueno, Vicente Martínez-Vizcaíno.

**Formal analysis:** Rubén Fernández-Rodríguez, Celia Alvarez-Bueno.

**Resources:** Rubén Fernández-Rodríguez.

**Software:** Sara Reina-Gutiérrez.

**Supervision:** Celia Alvarez-Bueno, Sara Reina-Gutiérrez, Sergio Nuñez de Arenas-Arroyo, Vicente Martínez-Vizcaíno.

**Validation:** Sara Reina-Gutiérrez, Ana Torres-Costoso, Sergio Nuñez de Arenas-Arroyo.

**Visualization:** Sara Reina-Gutiérrez, Ana Torres-Costoso, Sergio Nuñez de Arenas-Arroyo, Vicente Martínez-Vizcaíno.

**Writing – original draft:** Rubén Fernández-Rodríguez.

**Writing – review & editing:** Celia Alvarez-Bueno, Ana Torres-Costoso, Vicente Martínez-Vizcaíno.

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
