## [Decision Letter · Decision Letter 0]

7 Apr 2021

PONE-D-21-03751

Effectiveness of Pilates and Yoga to improve bone health in adult women: a systematic review and meta-analysis.

PLOS ONE

Dear Dr. Alvarez-Bueno,

Thank you for submitting your manuscript to PLOS ONE. After careful consideration, we feel that it has merit but does not fully meet PLOS ONE’s publication criteria as it currently stands. Therefore, we invite you to submit a revised version of the manuscript that addresses the points raised during the review process.

The requirements of the reviewers must be addressed accurately in order to consider the manuscript for publication.

We look forward to receiving your revised manuscript.

Kind regards,

Jose M. Moran

Academic Editor

PLOS ONE

Journal Requirements:

2. Please explain the reasons, and number of studies excluded for each reason, in the flow diagram. Thank you.

3. We note that the original search was performed in April 2020. Please discuss whether relevant literature has been published in the interim that would be expected to affect the results of the meta-analysis.

5. We note that this manuscript is a systematic review or meta-analysis; our author guidelines therefore require that you use PRISMA guidance to help improve reporting quality of this type of study. Please upload copies of the completed PRISMA checklist as Supporting Information with a file name “PRISMA checklist”.

Reviewers' comments:

Reviewer's Responses to Questions

**Comments to the Author**

1. Is the manuscript technically sound, and do the data support the conclusions?

Reviewer #1: No

Reviewer #2: Yes

2. Has the statistical analysis been performed appropriately and rigorously? 

Reviewer #1: No

Reviewer #2: Yes

3. Have the authors made all data underlying the findings in their manuscript fully available?

Reviewer #1: No

Reviewer #2: Yes

4. Is the manuscript presented in an intelligible fashion and written in standard English?

Reviewer #1: Yes

Reviewer #2: Yes

5. Review Comments to the Author

Reviewer #1: 1) First of all the title of the manuscript could be change to "Effectiveness of Pilates and Yoga to improve Bone Density in adult women" instead of Bone Health which carries a broader apprehension of musculo-skeletal well being. The only evaluation criterium used in the review is Bone Density.

2) To increase the practical value of the review, authors should consider separately:

a) Postmenopausal reports

b) Intensity of exercises

c) Differences between Yoga and Pilates

d) Short and long durations of exercises and

e) Other physical benefits

3) It might not be possible to include the additional data as proposed, authors could at least discuss about the variants, using information provided within or outside the systematic study.

Reviewer #2: In this meta-analysis, the authors investigate the effects of pilates and yoga on BMD in (mostly postmenopausal) women. They found no effect.

The work is adequately performed. It was registered with PROSPERO and reported according to PRISMA guidelines. I have the following specific comments:

1/ The conclusion should not necessarily be that "additional high-quality studies with an adequate intervention length are needed

to provide a more accurate picture of the evidence". I believe that yoga/Pilates is unlikely to change BMD, and the effect size here excludes a large effect. I suggest to remove this eternal "more evidence needed" mantra.

2/ Explain all abbreviations upon first use (e.g. ES, CI in the abstract)

3/ The authors should explain in the Introduction what exactly Pilates is and how it differs from Yoga.

6. PLOS authors have the option to publish the peer review history of their article (what does this mean?). If published, this will include your full peer review and any attached files.

Reviewer #1: No

Reviewer #2: **Yes: **Michaël R. Laurent

---

## [Author Response · Author response to Decision Letter 0]

14 Apr 2021

Reviewer #1: 

Reviewer’s comment:

1. First of all, the title of the manuscript could be change to "Effectiveness of Pilates and Yoga to improve Bone Density in adult women" instead of Bone Health which carries a broader apprehension of musculo-skeletal well-being. The only evaluation criterium used in the review is Bone Density.

Authors: We appreciate this reviewer’s suggestion. As suggested, we have modified the title of the manuscript: 

“Effectiveness of Pilates and Yoga to improve bone mineral density in adult women: a systematic review and meta-analysis”.

Reviewer’s comment:

2. To increase the practical value of the review, authors should consider separately:

a) Postmenopausal reports

b) Intensity of exercises

c) Differences between Yoga and Pilates

d) Short and long durations of exercises and 

e) Other physical benefits

Authors: We would like to thank the reviewer’s comment. We have carefully considered all the comments and incorporated them to the manuscript. 

a) We have conducted an additional subgroup meta-analysis based on the menopausal status of the participants, which are available in S8 and S9 Figs. We have properly modified methods, results, and discussion sections.

“Additionally, subgroup analyses based on (…) menopausal status as well as meta-regression(...).

“Lastly, meta-analyses for interventions vs CG and for the pre-post intervention analysis according to menopausal status are available in S8 and S9 Figs.”

b) Because of the impossibility to estimate the intensity of exercises, we have included this fact as a limitation of our study. 

“In this sense, it is supposed that high volume trainings lead to a smaller decrease of BMD in postmenopausal women (7) and also that the level of strain and body position during each exercise task may affect the load of the exercise impacting on BMD, (47) but these factors cannot be addressed in our study.”

c) The differences between Yoga and Pilates have been presented by subgroup analyses and discussed in discussion section. 

“As afore-mentioned and in line with our data, it seems that body position and physical demands during Pilates’ exercises produced more mechanical stress on bone when compared with Yoga exercises.” 

d) Aspects related with the duration of exercises have been addressed by subgroup based on the length, ≤ 24 weeks or >24 weeks, as well as meta-regression considering the weeks of the intervention. Method, result and discussion sections have been properly modified.

“Additionally, subgroup analyses based on the type of intervention (Pilates vs Yoga), length (≤ 24 weeks or >24 weeks) and menopausal status as well as meta-regression models by mean age, baseline BMD values after adjusting for height and baseline body mass index (BMI) and length were conducted to determine their potential effect on the pooled ES estimates.”

“The subgroup analysis based on the length, ≤ 24 weeks or >24 weeks, was conducted in the pre-post intervention showing a pooled ES of 0.13 (95% CI: 0.00 to 0.25; I2=40.5%) and 0.06 (95% CI: -0.07 to 0.19; I2=0.0%), respectively (S7 Fig).”

“(..) neither were the meta-regression models based on baseline BMD values after adjusting for height (p=0.36) or those based on BMI values (p=0.45) (S4 Table) or length of the intervention (p=0.57) (S10 Fig).”

“In this sense, it is well established that physical training should be maintained for at least one year to demonstrate substantial benefits in bone mass since the physiological cycle of bone remodelling lasts between four and six months,(48) and only two of the studies included in this review accomplished this.(31,36)”

e) Other physical benefits accomplished by mind-body exercises such as Pilates and Yoga have been discussed in the discussion section.

“Finally, because the type of exercise may modify the effect on BMD,(43) multicomponent strength and balance trainings have been recommended(39,44) for improving not only bone health, but also physical function in daily life activities, and for preventing falls and osteoporosis-related fractures associated with the decline of BMD”

Reviewer’s comment:

3. It might not be possible to include the additional data as proposed, authors could at least discuss about the variants, using information provided within or outside the systematic study.

Authors: Thank you for this suggestion. As it has not been possible to include some of the additional data proposed, we have discussed these items and included them as limitations of our study. 

Reviewer #2: 

In this meta-analysis, the authors investigate the effects of pilates and yoga on BMD in (mostly postmenopausal) women. They found no effect. The work is adequately performed. It was registered with PROSPERO and reported according to PRISMA guidelines. I have the following specific comments:

Reviewer’s comment:

1. The conclusion should not necessarily be that "additional high-quality studies with an adequate intervention length are needed to provide a more accurate picture of the evidence". I believe that yoga/Pilates is unlikely to change BMD, and the effect size here excludes a large effect. I suggest to remove this eternal "more evidence needed" mantra. 

Authors: We appreciate this recommendation. As suggested, we have modified the sentence in the Abstract section: 

“Despite of the non-significant results, the BMD maintenance in the postmenopausal population, when BMD detrimental is expected, could be understood as a positive result added to the beneficial impact of Pilates-Yoga in multiple fracture risk factors, including but not limited to, strength and balance.” 

Reviewer’s comment:

2. Explain all abbreviations upon first use (e.g., ES, CI in the abstract) 

Authors: We would like to apologize for this mistake. As suggested, it has been corrected. 

Reviewer’s comment:

3. The authors should explain in the Introduction what exactly Pilates is and how it differs from Yoga. 

Authors: We appreciate the reviewer’s suggestion. We have briefly described Pilates in the introduction section and the main differences with Yoga exercises. 

“Despite of the combined classification in Mind-body techniques, Pilates and Yoga present differences that may have influence on bone. For instance, Pilates is a therapeutic exercise highly focused on core-strengthening while Yoga is more related to breathing and meditation exercises.”

---

## [Decision Letter · Decision Letter 1]

26 Apr 2021

Effectiveness of Pilates and Yoga to improve bone health in adult women: a systematic review and meta-analysis.

PONE-D-21-03751R1

Dear Dr. Alvarez-Bueno,

We’re pleased to inform you that your manuscript has been judged scientifically suitable for publication and will be formally accepted for publication once it meets all outstanding technical requirements.

Kind regards,

Jose M. Moran

Academic Editor

PLOS ONE

Additional Editor Comments (optional):

Reviewers' comments:

Reviewer's Responses to Questions

**Comments to the Author**

1. If the authors have adequately addressed your comments raised in a previous round of review and you feel that this manuscript is now acceptable for publication, you may indicate that here to bypass the “Comments to the Author” section, enter your conflict of interest statement in the “Confidential to Editor” section, and submit your "Accept" recommendation.

Reviewer #1: All comments have been addressed

Reviewer #2: All comments have been addressed

2. Is the manuscript technically sound, and do the data support the conclusions?

Reviewer #1: Partly

Reviewer #2: Yes

3. Has the statistical analysis been performed appropriately and rigorously? 

Reviewer #1: N/A

Reviewer #2: Yes

4. Have the authors made all data underlying the findings in their manuscript fully available?

Reviewer #1: Yes

Reviewer #2: Yes

5. Is the manuscript presented in an intelligible fashion and written in standard English?

Reviewer #1: Yes

Reviewer #2: Yes

6. Review Comments to the Author

Reviewer #1: It is an interesting review but to draw any conclusion is probably unmeaningful. XXXXXXXXXXXXXXXXXXXXXXXXXXXXXXXXXXXXXXXXXXXXXXXXXXXXXXXXXXXXXXXXXXXXXXXXXXXXXXXXXXXXXXXXXXXXXXXXXXXXXXXXXXXXXXXXXXXXXXXXXXXXXXXXXXXXXXXXXXXXXXXx

Reviewer #2: The authors have sufficiently addressed all of my comments.

Requiring this answer to consist of more than 100 characters is a stupidity in the Plos Editorial Manager.

7. PLOS authors have the option to publish the peer review history of their article (what does this mean?). If published, this will include your full peer review and any attached files.

Reviewer #1: No

Reviewer #2: **Yes: **Michaël Laurent

---

## [Editor Report · Acceptance letter]

28 Apr 2021

PONE-D-21-03751R1 

Effectiveness of Pilates and Yoga to improve bone density in adult women: a systematic review and meta‑analysis 

Dear Dr. Alvarez-Bueno:

I'm pleased to inform you that your manuscript has been deemed suitable for publication in PLOS ONE. Congratulations! Your manuscript is now with our production department. 

Kind regards, 

on behalf of

Dr. Jose M. Moran 

Academic Editor

PLOS ONE